# Investigating the Free Volumes as Nanospaces in Human Stratum Corneum Lipid Bilayers Using Positron Annihilation Lifetime Spectroscopy (PALS)

**DOI:** 10.3390/ijms25126472

**Published:** 2024-06-12

**Authors:** Krystyna Mojsiewicz-Pieńkowska, Dagmara Bazar, Jacek Filipecki, Kordian Chamerski

**Affiliations:** 1Department of Physical Chemistry, Faculty of Pharmacy, Medical University of Gdańsk, Al. Gen. Józefa Hallera 107, 80-416 Gdańsk, Poland; dagmara-b@gumed.edu.pl; 2Institute of Physics, Faculty of Science & Technology, Jan Dlugosz University in Częstochowa, Armii Krajowej 13/15, 42-200 Częstochowa, Poland; j.filipecki@ujd.edu.pl (J.F.); kordian.chamerski@ujd.edu.pl (K.C.)

**Keywords:** free volume in stratum corneum, free volume in lipids bilayers, nanospaces in stratum corneum, positron annihilation lifetime spectroscopy, PALS technique, molecular properties of stratum corneum

## Abstract

This work is the first one that provides not only evidence for the existence of free volumes in the human stratum corneum but also focuses on comparing these experimental data, obtained through the unique positron annihilation lifetime spectroscopy (PALS) method, with theoretical values published in earlier works. The mean free volume of 0.269 nm was slightly lower than the theoretical value of 0.4 nm. The lifetime τ_3_ (1.83 ns with a coefficient of variation CV of 3.21%) is dependent on the size of open sites in the skin. This information was used to calculate the free volume radius R (0.269 nm with CV 2.14%), free volume size Vf (0.081 nm^3^ with CV 4.69%), and the intensity I_3_ (9.01% with CV 10.94%) to estimate the relative fractional free volume f_v_ (1.32 a.u. with CV 13.68%) in human skin ex vivo. The relation between the lifetime of o-Ps (τ_3_) and the radius of free volume (R) was formulated using the Tao–Eldrup model, which assumes spherical voids and applies to sites with radii smaller than 1 nm. The results indicate that PALS is a powerful tool for confirming the existence of free volumes and determining their size. The studies also focused on describing the probable locations of these nanospaces in SC lipid bilayers. According to the theory, these play an essential role in dynamic processes in biological systems, including the diffusion of low-molecular-weight hydrophobic and moderately hydrophilic molecules. The mechanism of their formation has been determined by the molecular dynamics of the lipid chains.

## 1. Introduction

Knowledge about the stratum corneum (SC) structure, with its important the molecular properties, continues to evolve. This enhances our grasp of skin barrier functions, including the mechanisms of the diffusion process. The achievements from this topic are useful for predicting the effectiveness and use of both dermal and transdermal drug delivery systems, as well as the barrier’s permeability to active pharmaceutical ingredients (APIs) [1,2]. The importance of that affects the direction of pharmaceutical technology development, including drug carriers that often utilize nature-inspired, biocompatible lipid nanostructures [3,4]. Estimating the skin barrier is also essential for ensuring the safe use of medicinal, care, and cosmetic products. This is important both for preventing undesirable substances from diffusing into the skin and for toxicological cases [5,6,7,8,9].

The efficacy of this barrier depends on a variety of closely interrelated factors. Key among them is the specificity of the integrated SC’s outermost skin layer, which is composed of several layers (approximately 15–20) of rigid corneocytes embedded in a lipid matrix, thus resulting in a tightly packed layer. The types of compounds involved—mainly lipids and proteins—as well as their molecular arrangement in different conformations, play a crucial role [4,10,11,12,13]. These elements determine the creation of spaces with sizes on both the micro and nano scales [1,12,13,14]. A crucial component of this effective barrier is the intercorneocyte multilayer lipid matrix (also known as multilamellar lipid membranes or lipid bilayers), with a thickness ranging from 20–35 nm [12,13,14].

Although penetrating the skin barrier is generally quite limited, substances with appropriate physicochemical properties have been shown to traverse the barrier through passive diffusion, in accordance with Fick’s Laws [6,8,14].

Importantly, substances can cross the skin barrier via several routes, and interfering with these structures, often increases undesirable diffusion [5,6,7,8,14,15,16]. This makes the area a particularly active focus of research. According to current understanding, compounds are transported into the skin through (1) the transcellular (intracellular) pathway by traversing the corneocytes; (2) the intercellular pathway by passing through the lipid bilayers that occupy the intercellular spaces surrounding the corneocytes; (3) the transappendageal pathway by crossing hair follicles, sebaceous glands, and sweat glands [12,13,14]; and (4) the canyons surrounding clusters of corneocytes [14,15,16,17].

It is worth emphasizing that canyons have been described only in a few publications to date. However, they have been identified as an important alternative pathway for the diffusion of substances into the skin, including the transport route for dermal formulations in nanoparticle form [14,15,16,17,18]. Canyons represent the lipid spaces located between clusters, thus reaching the dermoepidermal junction, which allows xenobiotics to diffuse directly into the blood or lymph vessels and bypass the SC barrier [14,15].

Apart from these pathways, some researchers have modeled skin permeability for both hydrophobic and hydrophilic solutes via four distinct SC routes, although they were described using different terms. Barry [18], Mitragotri [19,20,21], and subsequently Mercuri et al. [8], Neupane et al. [22], and Jankowskaya et al. (2021) [23]—who cited the first two authors—have postulated the following transportation pathways:Lateral diffusion along bilayers (K_p,lateral_), which is a dominant route for large lipophilic solutes;Diffusion through aqueous pores (K_p,pores_), which are waterfilled spaces within lipid bilayers created by imperfections in the lipid layer, and they are important for small and hydrophilic solutes;Diffusion through shunts (K_p,shunts_), such as sebaceous glands, hair follicles, and sweat glands that are the dominant pathways for large hydrophilic solutes; Diffusion through free volumes located in lipid bilayers (K_p,free volumes_), which are important for low-molecular-weight hydrophobic and moderately hydrophilic solutes. According to Mitragotri’s model, the permeability of the SC to a solute—whether hydrophilic or hydrophobic—can be mathematically represented by Equation (1). This equation expresses the sum of the permeabilities via four pathways: lateral diffusion along bilayers, diffusion through aqueous pores, diffusion through shunts, and free volume diffusion through SC lipid bilayers [19]:
K_p,SC_ = K_p,lateral_ + K_p,pores_ + K_p,shunts_ + K_p,free volumes_(1)

It should be emphasized that these transport pathways are not commonly cited in the literature and have not been thoroughly tested through experimentation. Many of them, such as the concept of free volumes, are primarily described through theoretical mathematical models. The idea of free volume in the SC was first considered in the context of molecular transport and the mathematical modeling of SC diffusivity by Potts and Guy in 1992, where it was considered as the molecular volume of the permeant [21,24]. This concept was further elaborated upon by Mitragotri [25], who provided additional details in subsequent works in the references [19,20,21].

According to these works, the free volume exists in lipid bilayers as pocket with a radius of 4 Å (0.4 nm) and is variously described as a free area, free spaces, or cavities [19,20,21,25]. Mitragotri focused on the contribution of free volume to the diffusion process, thus drawing upon Scaled Particle Theory (SPT). SPT is a statistical mechanical theory that calculates partition and diffusion coefficients based on the work required to create free volumes. This allows for the incorporation and movement of primarily hydrophobic and lipophilic solutes with a molecular weight (M) < 400 Da [19].

SPT also elucidates the mechanics of lipid chains, thus explaining that a solute can diffuse in all directions within the lipids. This includes transverse diffusion through lipid bilayers—perpendicular to the plane of the lipid head group—as well as lateral diffusion along lipid bilayers, which occurs in a plane parallel to that of the lipid head group [19,21,25].

The novelty of the present work lies in its provision of experimental data on free volumes and explaining the mechanism of their formation determined by the molecular dynamics of the lipid chains. Specifically, this study aims to investigate free volumes as nanospaces in the SC lipid bilayers using positron annihilation lifetime spectroscopy (PALS). Additionally, it evaluates the potential of this unique method for measuring human skin through statistical estimation. Beyond this, the study seeks to obtain experimental data through numerous repetitions using biological material sourced from both women and men aged 35–50 years. The collected data are then compared to Mitragotri’s theoretical value of free volume with a radius of 4 Å (0.4 nm) [19,20]. The work also aims to present the probable locations of these free volumes within the lipid bilayers.

Furthermore, considering that, to date, only two papers have been published that used the PALS method to determine free volume or pore sizes in the SC—one using animal skin from a Yucatan miniature pig [26] and the other using skin samples from just two human patients involving one with and one without basal cell carcinoma, as well as with and without squamous cell carcinoma [27]—our data are also compared and discussed with these earlier studies.

## 2. Results and Discussion

We have performed the experiments using the human skin ex vivo. Our studies proved that PALS, that has been used from many years to determine free volumes in polymeric materials, is a powerful tool that confirms the existence of free volumes or nanospaces in SC lipid bilayers and allows for the determination of their sizes.

In the first stage, we obtained the positron lifetime spectra, which contained various lifetime components. The observed three components that were fitted to the spectra, thus corresponding to the lifetime and intensity of τ_1_—p-Ps annihilation, the τ_2_—free positron annihilation responds of positron traps, and τ_3_—orthopositronium (o-Ps) annihilation reflecting the free volume spaces in which o-Ps annihilates. Therefore, due to the purpose of this work, only the longer-lived o-Ps components (τ_3_) were selected for the study.

To extract lifetimes and their corresponding intensities, the spectra were analyzed using the LT ver.9.0 fitting program. Based on the annihilation acts in the samples, we generated a positron lifetime spectrum. This spectrum represents a curve that describes the number of counts for all types of annihilation acts occurring in both the biological material being tested and the Kapton foil covering positron source, which was plotted as a function of time and expressed in nanoseconds [ns]. For each sample tested, 15 individual spectra were recorded. Figure 1 shows a typical spectrum depicting positron lifetimes [ns] in relation to the individual annihilation events recorded (counts).

The scheme of a PALS spectrum (Figure 1) depicts a third-lifetime component characterized by the lifetime of o-Ps. This lifetime is the inverse slope of the o-Ps component (slope = 1/τ_3_). This intensity (I_3_; I_o-Po_) is represented by the area under the respective o-Ps slope, with the formula for intensity being I_3_ = A_3_/τ_3_, which describes the probability of o-Ps formation relative to the total area under the spectrum [28]. The lifetime of o-Ps is attributed to the pick-off annihilation of the o-Ps, which occurs when the o-Ps is trapped in free volume sites within the sample. The third lifetime (τ_3_; ns) corresponds to the size of the free volume and is inversely related to the slope of the decay curve. The area I_3_ [%] is related to the concentration or amount of free volume sites (void sites) in human skin ex vivo. This lifetime (τ_3_; ns) is dependent on the size of the open sites in these biological materials and is used to calculate the free volume radius (R; nm), free volume size (V_f_; nm^3^), and, along with intensity I_3_ [%], to estimate the relative fractional free volume (f_v_; a.u.) in human skin ex vivo. The relationship between the lifetime of o-Ps and the radius of free volume (R) is formulated by the Tao–Eldrup model (Equation (3)), as presented in Section 3.4.1. This model presumes that these free volumes are spherical (Equation (4)) and applies to sites with radius smaller than 1 nm.

### 2.1. Evaluation of the Potential of the PALS Method for Examining Free Volumes in Human Skin Ex Vivo

The potential of the PALS method for examining free volumes in human skin ex vivo was confirmed based on the stability of measurements and the repeatability of results. Both the mean τ_3_ (Figure 2A) and the intensity I_3_ of long-lived orthopositronium lifetimes (Figure 2B) were determined. These parameters were derived from n = 15 measurement repetitions for each sample, which were conducted at intervals between 0 and 400 min. Specifically, measurements 1–15 were taken at the following times: 25, 50, 75, 100, 125, 150, 175, 200, 225, 250, 275, 300, 325, 350, and 375 min. A total of 11 independent ex vivo human skin samples, from men and women aged 35–50, were tested. Consequently, each measured (I_3_, *τ*_3_) or calculated (R, V_f_, and f_v_) parameter represents an average derived from a total of n = 165 replicates. Figure 2A,B show that the measurements for each biological sample (I to XI) remained stable over the analysis duration of 0–400 min. Moreover, the results for the entire collection of biological samples (n = 11) were reproducible. Interestingly, the variability in sample origin, including gender and age of the donors, did not significantly affect the scatter of the results. The use of human ex vivo skin of different origins for the study provided a compelling argument for the suitability of the PALS technique for skin testing. It should be noted that ex vivo human skin—whether obtained from autopsies as cadaver skin or sourced during surgery—is considered as the best predictive model for such studies. Therefore, it is commonly referred to as the gold standard in ex vivo conditions [22,29]. The study’s conclusions are particularly valuable given that researchers often have limited access to large skin samples from a single donor, which would guarantee the possibility of repeating the study multiple times to enhance statistical evaluation.

To assess the statistical stability of the measurements and the repeatability of the τ_3_ and I_3_ results, we conducted a Shapiro–Wilk test to evaluate if the data follow normal distribution. This test is particularly useful for smaller sample sizes. With a significance level set at α = 0.05, the null hypothesis (H_0_) was accepted, thus indicating that the τ_3_ measurement data follow a normal distribution, i.e., that most variables are close to the mean result. The analysis was performed using STATISTICA 13 software, and the results are presented in Figure 3. The statistical evaluation revealed a normal distribution for the measurements, supported by an obtained *p*-value of 0.07774, which is higher than the assumed significance level of α = 0.05. Although a slight left skewness was observed in the data, it did not affect the conclusion regarding the normality of the distribution, as this inference is primarily guided by the *p*-value.

In summary, the control samples are representative, which is a conclusion of significant relevance for current PALS studies. Specifically, this work focuses on the influence of cyclic and linear siloxanes on changes in the size of free volumes in the SC. Such findings could potentially advance studies investigating the interactions between various substances and skin at the nanostructure level.

To proceed with this type of research, it is necessary to compare changes in free volume sizes relative to a representative group of control samples.

### 2.2. Experimental Data

The main purpose of this work was to experimentally investigate free volumes as nanospaces in human skin, specifically within the SC lipid bilayers, and to compare these findings with theoretical values presented by Mitragotri [19,20]. Moreover, only two papers have been published so far in which the authors used the PALS method to determine the free volume or pore sizes in the SC. In the first case, an animal skin, specifically from Yucatan miniature pigs, was used [26], while the second case involved human skin samples from two patients, thus comparing skin both with and without basal cell carcinoma, as well as with and without squamous cell carcinoma [27]. Given this limited existing research, our data are especially valuable, and we have taken care to compare and discuss our findings in relation to these prior studies. The measured experimental parameters include the o-Ps lifetime (τ_3_) and intensity (I_3_), which were obtained through PALS. Calculated values such as free volume radius (R), free volume size (V_f_), and fractional free volume (f_v_) for human skin ex vivo are summarized in Table 1. The o-Ps lifetime (τ_3_) is pivotal, as it confirms the presence of free volumes and enables the calculation of their radius. The o-Ps intensity (I_3_) relates to the probability of o-Ps formation and the number of free volume spaces.

Based on our PALS measurements, we obtained a mean value of τ_3_ = 1.83 ns with a coefficient of variation (CV) of 3.21%. This is consistent with values presented by Itoh et al. for animal skin τ_3_ = 2.1 ns [26] and by the Jean’s research group for human skin (without carcinoma) τ_3_ = 2.043 and 2.081 ns [27]. These similarities suggest a comparable o-Ps annihilation phenomenon and, consequently, a similar free volume radius (R). Our data showed an o-Ps intensity I_3_ = 9.01%, with an RSD = 10.94%. This implies an insignificant scatter in the results related to the concentration of free volume sites, thus further confirming the limited variability in our skin samples. In addition, the obtained I_3_ value is slightly lower than the 14.65% and 14.16% reported by the Jean research group [27].

Our study proved the presence of free volume in human skin, thus demonstrating that the size of these nanospaces is on the atomic scale. Due to the fact that the obtained results are compared with the theoretical values, as well as experimental works involving nonunified units, all values are presented in two units: [nm] and [Å]. The mean value of the measured free volume radius (R) was 0.269 nm (or 2.69 Å), with a CV of 2.14%. This aligns well with the values of 2.887 and 2.921 Å published by the Jean’s research group [27], although these data are lower than the theoretical value of 0.4 nm (4 Å) suggested by Mitragotri [19,20].

Similarly, as shown in Table 1, the calculated free volume size of 0.081 nm^3^ (81.5 Å^3^), with a CV of 4.69%, was obtained from the base equation V_f_ = 4/3 π R^3^ (Equation (4)), which correlates with the values of 100.8 and 104.4 Å reported by the Jean’s research group [27,30,31]. Apart from these, we observed that the values of free volume radius (R) and free volume size (V_f_) in human skin are comparable to those found in animal skin studies [26]. According to Itoh et al. [26], the Yucatan miniature pig stratum layer had a diameter of at least 0.59 nm, which translates to a radius R = 0.295 nm and a free volume size V_f_ = 0.098 nm^3^.

Another interesting parameter is the fractional free volume (f_v_), which is calculated using the equation $f_{v}=CV_{f}I_{3}$ (Equation (5)), where C = 0.0018 is an empirical dimensionless parameter. The fractional free volume is defined as the ratio of the specific free volume to the specific volume of a material, such as a polymer, biopolymer, or lipid bilayer [32,33]. In our study, the mean value of the fractional free volume was 1.32, with a CV of 13.68%. This is consistent with the values of 2.66 and 1.48 for human skin, as reported by the Jean’s research group [27].

To sum up, using the unique PALS method, this work provides experimental evidence for the existence of free volumes in the human SC. Although there are several mathematical models for calculating this skin nanospaces, PALS emerges as the preferred choice for obtaining experimental data. The results presented in this paper align with experimental results for animal skin [26] and cancerous skin [27,30,31], although they are lower than theoretical values [19,20]. It should be emphasized that if no free volumes were present in the biological material studied, no o-Ps would be formed, i.e., no third component *τ*_3_ would be obtained [34,35].

### 2.3. Molecular Mechanism of Creating Free Volumes in Stratum Corneum

The concept of the free volume existing in human biological structures has already been a topic of discussion among researchers. Initially, free volumes in the phospholipid bilayers of biological membranes were reported [21,36] and subsequently extended to include the SC [26], thus incorporating lipid bilayers [21]. While the existence of free volumes in human skin has been acknowledged in several publications [1,19,20,21,25,26,27], their hypothetical locations have not yet been described in detail. Itoh et al. (2008) used the PALS technique to determine the free volume in Yucatan miniature pig SC (with a diameter of 0.59 nm and a radius of 0.295 nm), but they referred to it as a “pore”, which could be misleading [26]. In conclusion, they were unable to specify whether these free volumes exist in corneocytes and/or the intercellular lipid matrix [26].

However, some researchers have suggested that free volumes are located in the unique molecular organization of the SC lipid bilayers [1,19,20,21,25,26,27]. For instance, Cevc et al. claimed that free volume is formed between the crystalline lipid lamellae, where the fluid region exists with less ordered lipids and more flexible hydrophobic chains [1]. Moreover, they concluded that the fluid lipids in skin barrier are crucially important for the transepidermal permeation of the lipophilic and amphiphilic molecules, as they provide the “free volume” needed for the insertion and diffusion of such molecules [1]. Similar to Cevc et al. inference [1], Mitragotri et al. also confirmed that free volumes are localized in these microstructures near the center [19,21,25].

Hence, considering the achievements of scientists to date, which are summarized in recent publications from 2019–2023 [37,38,39,40,41,42,43], we present the actual state of knowledge, which makes it possible to formulate the hypothesis and probably indicate the most active locations for free volume formation relevant for skin diffusion.

Firstly, the SC lipid bilayers are composed of ceramides (CERs), cholesterol (CHOL), and free fatty acids (FFAs), presented in Figure 4A–C, and involve two coexisting lamellar phases with repeating distances of approximately 6 and 13 nm, which are referred to as the short periodicity phase (SPP) and long periodicity phase (LPP), respectively, as shown in Figure 5A,B [37]. Although ceramides have a diverse structure (approximately 25 CER subclasses have been identified to date), many different models have been investigated, thus focusing on the key role of the ceramide type in the formation of SPPs and LPPs. Based on the current conclusions [37,38,39,40,41,42,43], in this work a simplified model was adopted, in which CERs are in a linear, not hairpin conformation, and the sphingosine chain is located next to CHOL, while the FFAs are adjacent to the acyl chain of the CERs (Figure 5A,B). The SPP is mainly formed by N-lignoceroyl sphingosine (Cer[NS]), whereas the LPP formation is determined by the presence of N-(32-linoleoyloxy)dotriacontanoyl sphingosine (Cer[EOS]) localized in the outer layers, as well as Cer[NS] and N-lignoceroyl phytosphingosine Cer[NP] in the inner layers [37,39].

An important consideration in formulating our hypothesis regarding the likely location of free volumes was the research of Hasler’s and Vavrova’s groups [39,40]. The authors reported that both Cer[NS] and Cer[EOS] contain the dynamic regions, as shown in the Figure 4B,C. In the case of Cer[NS] (Figure 4B), dynamics regions 1 and 3 exist, where region 1 depends on the sphingosine moiety (sphingosine chain) creating the fluid phase with 5% isotropic terminal four carbons, whereas region 3 located at the end of the acyl chain forms an isotropic phase with dynamic character [39,40]. In the central part of Cer[NS], a rigid crystalline phase exists, which is outlined as region 2. 

Similar to Cer[NS], the ultralong N-acyl chain of Cer[EOS] (Figure 4C) exhibits surprising molecular dynamics (region 1 and even highly dynamic region 3) with increased isotropic motion toward the isotropic ester-bound linoleate [39,40]. These researchers also discovered that the LPP structure contains alternating fluid (sphingosine chain-rich), rigid (acyl chain-rich), isotropic (linoleate-rich), rigid (acyl-chain-ich), and fluid layers (sphingosine chain-rich) [39].

Due to the fact free volumes require flexible and mobile lipid chains to be formed, therefore, according to our hypothesis, the locations are visible among the fluid and isotropic phases in both the SPP and LPP arrangements (Figure 5A,B). Moreover, the presence of the Cer[EOS] in the LPP with increased isotropic motion due to the isotropic ester-bound linoleate may be the reason that more free volumes occur in this part, as shown in Figure 5B.

## 3. Materials and Methods

### 3.1. Reagents

A 0.9% NaCl solution was prepared to rinse and cleanse the skin samples both before preliminary preparation and after the removal of subcutaneous fat. The solution consisted of sodium chloride (Polish Chemical Reagents, Gliwice, Poland) and purified water (Elix 3 System, Millipore, Bedford, MA, USA). To remove sebum from the skin surface, chloroform was used, which was also purchased from Polish Chemical Reagents in Gliwice, Poland.

### 3.2. Human Ex Vivo Skin as a Biological Material to Study

In this study, frozen–thawed samples of full thickness human ex vivo skin were used, which were obtained from cadavers during necropsy. After collecting samples from the abdominal part of the body (15–20 cm in length, 2–3 cm in width) within 48 h of death (males and females aged 35–50 years), a preliminary skin preparation was performed. Due to the use of biological material, the study received approval from the Independent Bioethics Commission for Research at the Medical University of Gdańsk (nos. NKBBN/309/2013 and NKBBN/449/2020) and was conducted in accordance with the guidelines defined by the OECD [44,45] and WHO [46].

### 3.3. Preparation of Human Ex Vivo Skin

The subcutaneous tissue was carefully separated from the dermis layer using scissors. In the next step, the skin was cut into smaller pieces (2 cm in length, 1 cm in width) and rinsed with a 0.9% NaCl solution. Dried samples of skin, with an average full thickness of 1.40 ± 0.2 mm (comprising the SC, epidermis, and dermis layers), were wrapped in aluminum foil and stored at −20 °C. Before use in the experiment, the samples were thawed at room temperature (Figure 6(1A)).

We demonstrated in our team’s earlier work [47] that the freezing process has no impact on the skin’s barrier properties; therefore, the use of frozen–thawed samples of the biological material did not affect the results of this research. The integrity of the skin samples’ barrier was confirmed visually using a magnifier with 4-fold magnification to examine the upper side of the skin. Additionally, the integrity of each thawed skin sample’s surface and the maintenance of their barrier properties were verified using the Trans-Epithelial Electrical Resistance technique (TEER), as shown in Figure 6(1B). For this, a Millicell ERS-2 (Millipore, USA) data bridge connected to two stainless steel electrodes was used. TEER was expressed in kΩ for the exposed skin surface area (0.65 cm^2^). All performed measurements demonstrated values above the required limit of 2 kΩ/cm^2^ (average 5 kΩ/cm^2^), thus confirming the integrity of the tested samples [48,49,50].

Subsequently, the skin fragments were placed in a chamber for 24 h. The chamber consisted of Petri dishes with the bottom covered by blotting paper soaked in distilled water and a piece of aluminum foil placed on the blotting paper in a manner that did not fully cover it (Figure 6(1C)). This setup maintained reproducible moisture conditions for each skin sample. Sebum was removed from the skin surface by gently wiping it with a cotton swab soaked in a small amount of chloroform.

### 3.4. Positron Annihilation Lifetime Spectroscopy Technique

#### 3.4.1. Theoretical Aspect of the PALS

The PALS technique is presented in Figure 7. In PALS measurements, the positron plays a key role. The positron (e^+^) is an antiparticle of the electron (e^−^), thereby possessing the same mass but an opposite charge. Positrons are formed as a result of the spontaneous β^+^ decay of the unstable atomic nuclei of certain radioactive isotopes. Positron annihilation refers to the process in which the entire mass of both the positron and the electron, as well as their kinetic energy, are converted into the energy of photons of electromagnetic radiation according to Equation (2) [51]:(2)$$\sum_{i}h\nu_{i}=2m_{e}c^{2}+E_{-}+E_{+}$$
where $\sum_{i}h\nu_{i}$ is the energy of emerging photons, m_e_ is the rest mass of the electron and positron, c is the speed of light, and E_+_ and E_−_ are the kinetic energies of the positron and electron.

Therefore, the examination of photons generated in the annihilation process provides information on the state of the annihilating electron–positron pair. Annihilation is only possible if it adheres to all conservation laws, specifically the conservation of energy, momentum, angular momentum, charge, and parity. During annihilation, the emission of either an even (2γ) or odd (3γ) number of gamma quanta is possible.

In addition to free annihilation, the formation and annihilation of a positron–electron bound state, known as a positronium (Ps), also occurs [51]. The annihilation of a positron, with a maximum kinetic energy of about 545 keV, is preceded by thermalization within the material. This involves a quick loss of positron energy due to scattering and excitation of the medium. Upon reaching the thermal vibrational energy of the lattice, the positron begins to move through the structure by diffusion, thus interacting with structural inhomogeneities and free electrons to form the positronium.

Two different types of positroniums can be distinguished based on the alignment of spins: parapositronium (p-Ps) with antiparallel spins (resulting in 2 gamma quanta annihilation and a 0.125 ns lifetime in a vacuum), and orthopositronium (o-Ps) with parallel spins (resulting in 3 gamma quanta annihilation and a 142 ns lifetime in a vacuum). One-quarter of the total Ps yield appears as p-Ps (singlet state) and three-quarters as o-Ps (triplet state).

The physical properties of positronium can change due to its interaction with the surrounding medium. One observed phenomenon is the shortening of the three-photon o-Ps mean annihilation lifetime, also known as pick-off annihilation [51,52]. This involves the ability of the positron, which is part of the o-Ps, to perform two-photon annihilation with one of the electrons in the Ps’ immediate surroundings [53,54].

The presence of free volume—an area with zero electron density—with the diameter of no less than 0.106 nm (1.06 Å) is necessary for Ps to exist in a condensed medium. Local free volumes occur in materials due to irregular molecular packing. The Tao–Eldrup model has been used to describe the relationship between the o-Ps lifetime and the size of free volume. The model has fully accomplished its task, and it has been used for years to calculate the dimensions of free volume holes in materials by numerous research centers all over the world. It assumes that the Ps, trapped within a spherical free volume, can undergo spontaneous annihilation, thus emitting either three quanta in the regular process or two quanta as a result of pick-off annihilation [55,56,57,58,59,60,61,62].

The Ps localized in free volume spaces serve as an indicator of their mean radius R in terms of long-lived τ_3_ lifetimes. The relative intensity of this component I_3_ correlates with the density of available Ps decay sites in accordance with the Tao–Eldrup Equation (3):(3)$$\tau_{3}\left(ns\right)=0.5{\left[1-\frac{R}{R+\Delta R}+\frac{1}{2\pi }sin\left(\frac{2\pi R}{R+\Delta R}\right)\right]}^{-1}$$
where ΔR = 1.656 Å is the thickness of the layer of electrons on the free volume wall, from which the o-Ps annihilate during the pick-off process.

Considering the spherical shape, the mean size of free volume can be expressed as follows:(4)$$V_{f}=\frac{4}{3}\pi R^{3}$$

The free volume size can then be used to calculate the fractional free volume from the following formula:(5)$$f_{v}=CV_{f}I_{3}$$
where C is the parameter to be calibrated by other measurements. The constant is typically equal to 0.001–0.002 [51,56].

#### 3.4.2. PALS Spectra Measurements

A schematic representation of the research methodology is shown in Figure 6. Detailed descriptions of the ex vivo skin sample preparation procedures are presented in Figure 6(1A–1C) and described in Section 3.2 and Section 3.3. The experimental setup for a single measurement required a “sandwich” system (Figure 6(2C)) comprising two stacks of skin samples with identical dimensions. These samples were collected from the same donor and positioned with the SC layer facing the positron source. The samples were placed on nickel plates. A ^22^Na isotope positron source with an activity of 3.5 × l0^5^ Bq was centrally positioned between two pieces of 8 µm thick Kapton foil (DuPont™, Circleville, OH, USA), which were held together by a metal rim.

To construct this “sandwich” system, the first skin sample (no. 1) was taken from the chamber (as described in Section 3.3, Figure 6(1C)). It was placed on a nickel plate so that the SC layer was on top, and the dermis was in contact with the nickel plate. Then, skin sample no. 1, along with the nickel plate, was placed on the lower scintillation counter so that the nickel plate touched the surface of the lower scintillation counter. The upper surface of skin sample no. 1 was protected with a piece of Kapton film, which was gently laid on sample no. 1 using tweezers (Figure 6(2A)).

In the next step, a positron-emitting source was precisely placed on the surface of the formed system, including sample no. 1, in such a way that the fragment containing the radionuclide was located in the central part of skin sample 1. Next, another Kapton film was laid over the positron source. Skin sample no. 2, along with a nickel plate prepared in the same manner as for skin sample no. 1, was precisely placed with tweezers. The SC layer faced the positron source and touched the Kapton film (Figure 6(2B)).

It was important to ensure that sample no. 2 fully covered the positron source and overlapped with sample no. 1. This arrangement minimized the annihilation of positrons outside of the sample array. The completed “sandwich” system was then carefully pressed using the upper scintillation counter. Figure 6(2C) shows the system before this final pressing step.

The experimental PALS spectra were acquired using an ORTEC “start-stop” system [63], which incorporated two scintillating counters (Scionix, Holland BV, Arnhem, The Netherlands) equipped with BaF_2_ scintillators and ORTEC^®^ electronics (ORTEC, Oak Ridge, TN, USA). To ensure the reliability of the measurements, the PALS spectra were recorded under controlled conditions: at room temperature (*T* = 22 °C) and a relative humidity of 35%. The measurements were performed using normal measurement statistics involving 10^6^ counts under the PALS curve. The channel width was set at 6.5 ps, thus allowing for a total of 8192 channels. Data from the PALS spectra were processed using the LT ver.9.0 program [64], and the average lifetime was stabilized in accordance with Equation (6) [52,53]:(6)$$\tau_{av}=\sum_{i=1}^{k+1}I_{i}\tau_{i}$$
where τ_i_ and I_i_ denote the positron lifetime and intensity of the corresponding ith fitting component.

## 4. Conclusions

The literature review over the past decade offers limited information on the size and concept of the creation the free volumes. Hence, this research expands the knowledge of the molecular properties of the stratum corneum and explains the mechanism of their formation determined by the molecular dynamics of the lipid chains. The solute diffusion is hypothesized to involve jumps between free volume pockets that open due to density fluctuations in the continuously dynamic lipid chains [19,25]. This topic is important for various research group for understanding how this nanospace impacts on various aspects, particularly on the diffusion process of molecules through stratum corneum and deeper skin layers where the blood and lymphatic vessels are located. This is especially important for low-molecular-weight hydrophobic solutes and moderately hydrophilic solutes.

In conclusion, a deeper understanding of the mechanisms of substance transport through the skin is crucial for the development of new minimally invasive transdermal therapies. It also aids to predict the efficacy of dermal and transdermal drug delivery systems and ensures the safe use of medicinal, personal care, and cosmetic products.

## Figures and Tables

**Figure 1 ijms-25-06472-f001:**
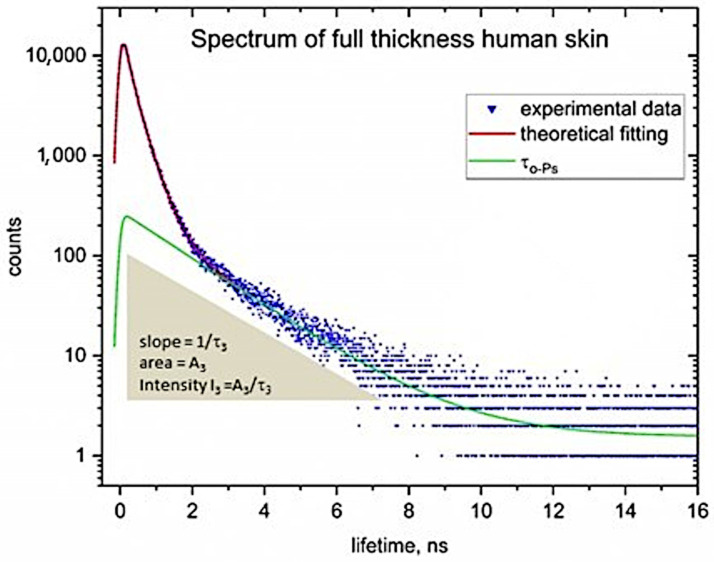
Exemplary positron lifetime spectrum received for full thickness human skin ex vivo (blue data points) with the fitted theoretical spectrum (red line) and a component of an o-Ps lifetime (green line).

**Figure 2 ijms-25-06472-f002:**
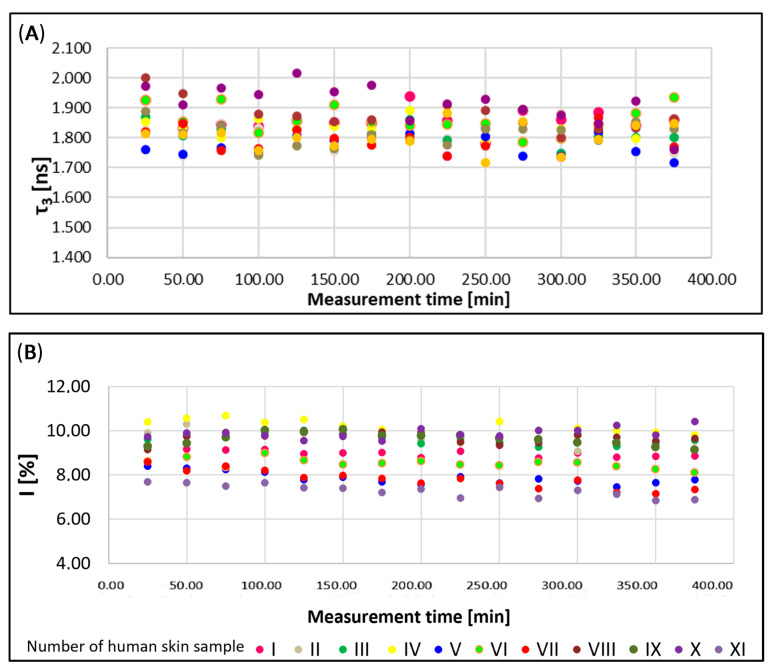
Course of measurements of the mean orthoposition lifetime τ_3_ [ns] as a function of the analysis duration 0–375 min (**A**) and the course of measurements of I_3_ [%] intensity at 15 time points from 0 to 375 min (**B**).

**Figure 3 ijms-25-06472-f003:**
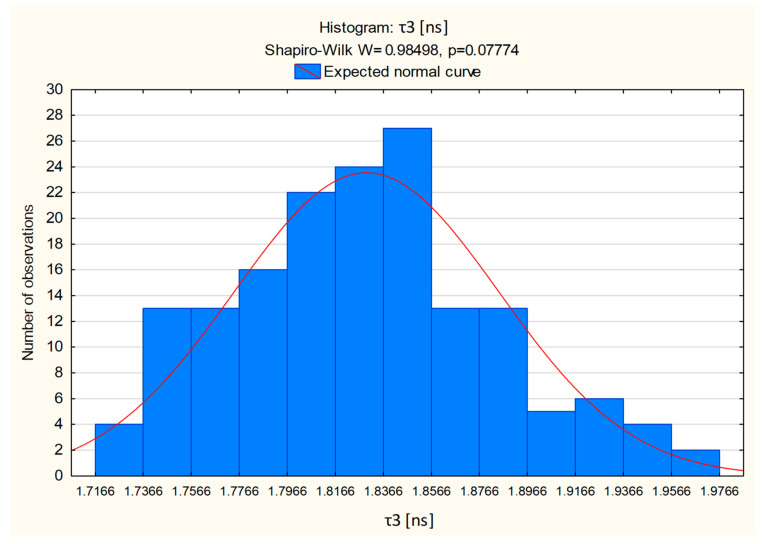
Histogram for τ_3_ [ns] using Shapiro–Wilk normality test.

**Figure 4 ijms-25-06472-f004:**
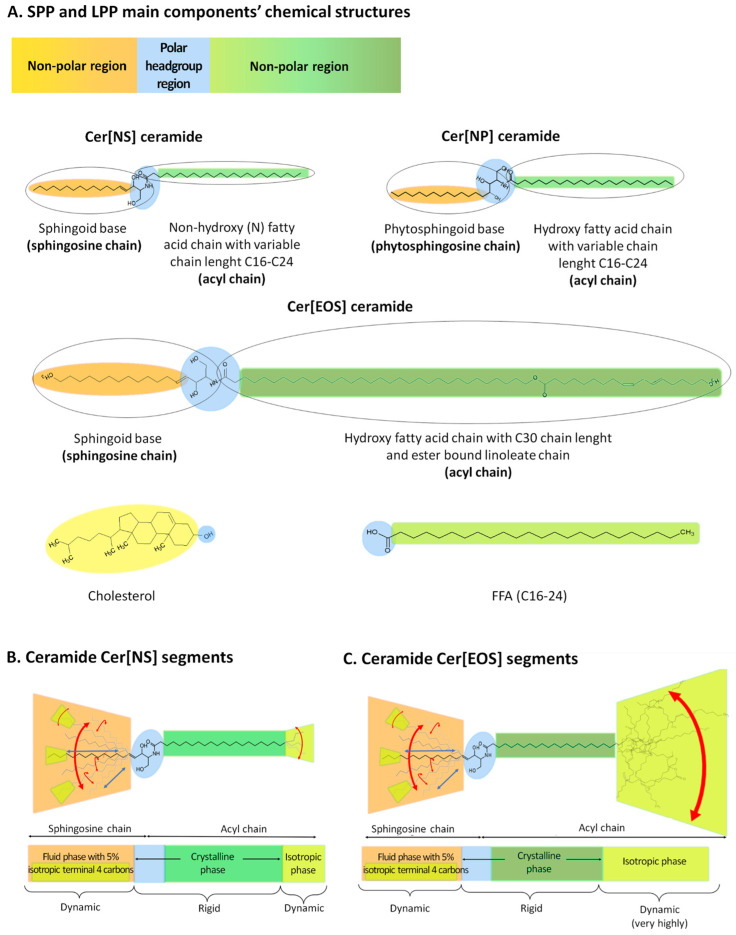
Chemical structures of Cer[NS], Cer[NP], and Cer[EOS], as well as ceramides, cholesterol, and FFAs. The components of two lamellar phases coexisting in SC lipid bilayers: the short periodicity phase (SPP) and long periodicity phase (LPP). (**A**). Segments with different mobility of the main components in these phases (different arrows): Cer[NS] (in SPP) (**B**) and Cer[EOS] (in LPP) ceramides (**C**).

**Figure 5 ijms-25-06472-f005:**
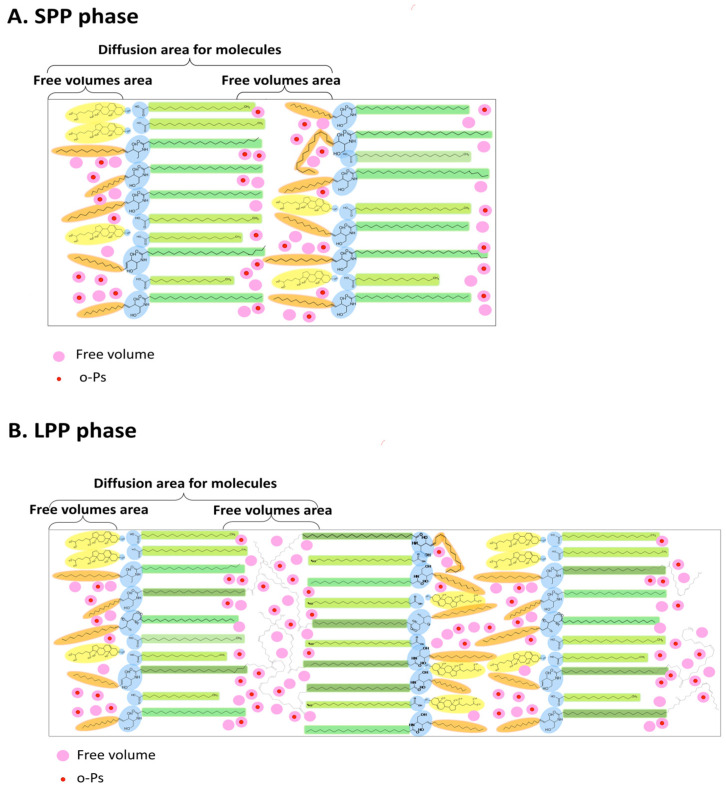
The SPP and LPP models with a hypothetical location of free volumes; orthopositronium o-Ps, trapped within a spherical free volume, can undergo spontaneous annihilation, thus emitting two quanta as a result of pick-off annihilation, as measured by PALS method.

**Figure 6 ijms-25-06472-f006:**
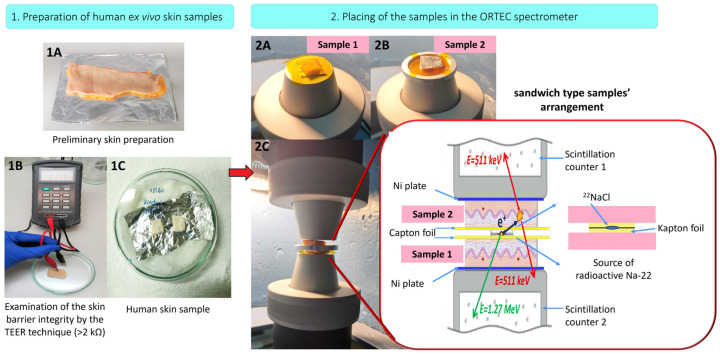
Research methodology: stage 1—preparation of the human ex vivo skin samples (**1A**,**1B**,**1C**); stage 2—placing of the samples in the PALS spectrometer (**2A**,**2B**,**2C**).

**Figure 7 ijms-25-06472-f007:**
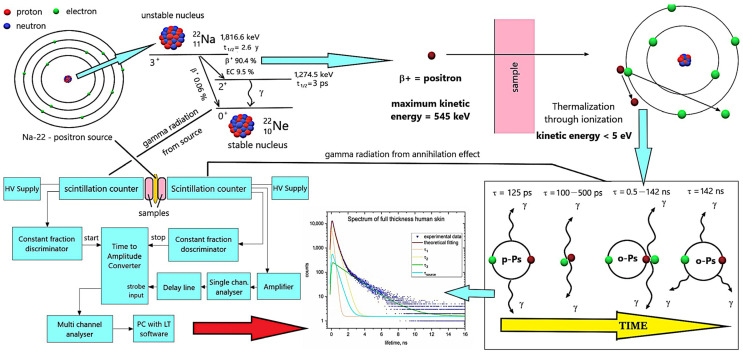
The positron annihilation lifetime spectroscopy (PALS) technique principle.

**Table 1 ijms-25-06472-t001:** Analyzed PAL spectra data for ex vivo human skin.

Sample	Measured Parameters		Calculated Parameters		
	o-Ps Lifetime [ns]	Intensity [%]	Free Volume Radius [nm]	Free Volume Size [nm^3^]	Fractional Free Volume [a.u.]
	τ_3_	I_3_	R	Vf	fv
I	1.86	8.96	0.272	0.084	0.00135
II	1.80	9.66	0.265	0.078	0.00136
III	1.82	9.65	0.268	0.080	0.00139
IV	1.84	10.18	0.269	0.082	0.00150
V	1.77	7.86	0.263	0.076	0.00108
VI	1.86	8.52	0.272	0.084	0.00129
VII	1.80	7.79	0.265	0.078	0.00109
VIII	1.87	9.68	0.273	0.085	0.00148
IX	1.81	9.62	0.267	0.080	0.00139
X	1.92	9.89	0.277	0.089	0.00158
XI	1.80	7.29	0.266	0.079	0.00104
Mean value	1.83	9.01	0.269	0.081	0.00132
CV [%]	3.21	10.94	2.14	4.69	13.68

Number of human skin ex vivo (female, male, age: 35–50); n = 11. Number of measure repetitions for each sample in the time range 0–400 min; n = 15. Mean values originate from repetitions of measures; n = 11 × 15 = 165. CV—coefficient of variation [%].

## Data Availability

The data are available from the corresponding authors upon reasonable request.

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
