# Peer review of "Investigating the Free Volumes as Nanospaces in Human Stratum Corneum Lipid Bilayers Using Positron Annihilation Lifetime Spectroscopy (PALS)"

_ijms, 2024, doi:10.3390/ijms25126472_

Round 1

Reviewer 1 Report

Comments and Suggestions for Authors

This paper provides a novel approach to monitor free volume in multilamellar stacks of lipid bilayers, which finds biological relevance in skin permeation processes. 

The manuscript is well organized and the technique and related results well explained, however, it requires grammatical revision by a native English speaker.

Several points need to be clarified before this paper can be accepted:

1) The authors should clarify whether free volume is restricted to interbilayer volume or also also intrabilayer volume (free volume between the lealflets) can be considered. If that is the case, they should stress that free volume is restricted to the study of multilayer films and multilamellar vesicles

2) What is the state of the art for free volume measurements? The authors should include a paragraph about other techniques capable of quantifiying free volume and state advantages and disadvantages compared to PALS

3) A distinction between average volume per molecule and free volume should be made. For the community of scientists working in lipid bilayer thermodynamics this difference should be clear. The average volume occupied by a molecule in a lipid bilayer can be estimated  from the specific volume for multilamellar vesicles samples using vibrating tube densitometry, see, i.e., L. Bar et al., Stability of supported hybrid lipid bilayers on chemically and topographically-modified surfaces, Colloids Surfaces A Physicochemical and Engineering Aspects 664, 131125 (2023).

4) Uncertainty in the values displayed in Table 1 should be included

Comments on the Quality of English Language

Revision by native speaker

Reviewer 2 Report

Comments and Suggestions for Authors

The manuscript "Investigating the free volumes as nanospaces in human stratum corneum lipid bilayers using positron annihilation lifetime spectroscopy (PALS)" describes thorough PALS measurements on several natural skin samples and discusses the scientific relations. Finally, only the voids inside the skin samples are characterized, while many scientific issues are discussed in this context without a real connection to their characterization. I would promote the manuscript for publication after some tighter connections are made.

The authors discuss the diffusion constants Kxy and that this is the real aim. Can't there be given some example values and maybe be explanations how K relates to the void size and concentration?? The connection in the actual manuscript is so loose. Why mentioning it at all?

The concentration of SPP and LPP lipid formations is also discussed in theory. What is the realistic frequency in real skin. I could be measured by SAXS for instance. Again, what would be the relation to an effective K if there is more SPP or LPP? This is all theory and the connection to reality is missing.

Comments on the Quality of English Language

Please double check spelling errors.

I only detected the superflous "the" in the first sentence "The knowledge about the stratum corneum (SC) structure with its important the molecular properties, continues to evolve."

Reviewer 3 Report

Comments and Suggestions for Authors

The molecular properties of the stratum corneum have been elaborated here. It reviews the mechanism of the formation of this layer involving the molecular dynamics of the lipid chains. The existence of free volumes in human stratum corneum has been proven here.

Using the positron annihilation lifetime spectroscopy (PALS) method data were produced and were compared with available theoretical data. Both are found compatible with each other. The study concludes that the solute diffusion may involve jumps between free volume pockets that open due to density fluctuations in the continuously dynamic lipid chains. The claims are trustable based on the pieces of evidence (data and plots) provided.

The article is well written.  The materials and Methods sections are properly documented. Results are processed properly and readily available for understanding.

It's a positive note from me. Good luck!

Comments on the Quality of English Language

Minor editing is required.
